# RAPPER: REINFORCED RATIONALE-PROMPTED PARADIGM FOR NATURAL LANGUAGE EXPLANATION IN VISUAL QUESTION ANSWERING

**Kai-Po Chang**[1]    **Chi-Pin Huang**[1]    **Wei-Yuan Cheng**[1]    **Fu-En Yang**[1,2]
**Chien-Yi Wang**[2]    **Yung-Hsuan Lai**[1]    **Yu-Chiang Frank Wang**[1,2]

[1]Graduate Institute of Communication Engineering, National Taiwan University, Taiwan
[2]NVIDIA, Taiwan
`{r11942093,r11942097,b08502072,r10942097}@ntu.edu.tw,`
`{fredy,frankwang,chienyiw}@nvidia.com`

## ABSTRACT

Natural Language Explanation (NLE) in vision and language tasks aims to provide human-understandable explanations for the associated decision-making process. In practice, one might encounter explanations which lack informativeness or contradict visual-grounded facts, known as *implausibility* and *hallucination* problems, respectively. To tackle these challenging issues, we consider the task of visual question answering (VQA) and introduce *Rapper*, a two-stage **R**einforced R**a**tionale-**P**rom**p**t**e**d Pa**r**adigm. By knowledge distillation, the former stage of *Rapper* infuses rationale-prompting via large language models (LLMs), encouraging the rationales supported by language-based facts. As for the latter stage, a unique Reinforcement Learning from NLE Feedback (RLNF) is introduced for injecting visual facts into NLE generation. Finally, quantitative and qualitative experiments on two VL-NLE benchmarks show that *Rapper* surpasses state-of-the-art VQA-NLE methods while providing plausible and faithful NLE.

## 1 INTRODUCTION

Deep learning has achieved remarkable success in vision-language (VL) tasks such as visual reasoning (Suhr et al., 2017), visual question answering (VQA, Goyal et al., 2017), and visual entailment (Xie et al., 2019). Take VQA as an example, while these models exhibit impressive ability in inferring answer descriptions from the given image-question pairs, its decision-making process remains an unsolved problem. As a result, such a black-box manner severely restricts their applicability in certain real-world scenarios (e.g., medical VQA, Lin et al., 2023), where the interpretability of the learning model is crucial for establishing trustworthy systems. To tackle this long-standing challenge, some approaches adopt attention mechanisms (Anderson et al., 2018) or gradient-based activations (Selvaraju et al., 2017), focusing on highlighting image regions which are relevant to the associated prediction. However, such visual explanations might not be desirable for VL tasks (e.g., those beyond classification) due to the lack of reasoning process (Kayser et al., 2021; Sammani et al., 2022). As a result, Natural Language Explanation (NLE) has emerged as a potential alternative, which aims to interpret the underlying reasoning process by natural language descriptions.

To extend NLE for vision-language tasks (i.e., VL-NLE), Park et al. (2018) and Kayser et al. (2021) introduced the benchmarks for explaining the decision-making process with NLEs for VQA and visual entailment tasks, respectively. Subsequent VL-NLE works have evolved into two research lines. The first research line (Park et al., 2018; Marasović et al., 2020) focuses on how to improve their pipeline from an architecture perspective for training NLE generators within a fully supervised learning manner. On the other hand, Sammani et al. (2022) and Suo et al. (2023) emphasize the utilization of unlabeled pre-training data to enhance the language models' NLE capability.

Despite significant advancements, most existing VL-NLE works require training in a full supervised manner. They might encounter problems where the explanations are irrelevant to the questions or contradictory to the established supporting facts (Majumder et al., 2021). The other potential concern

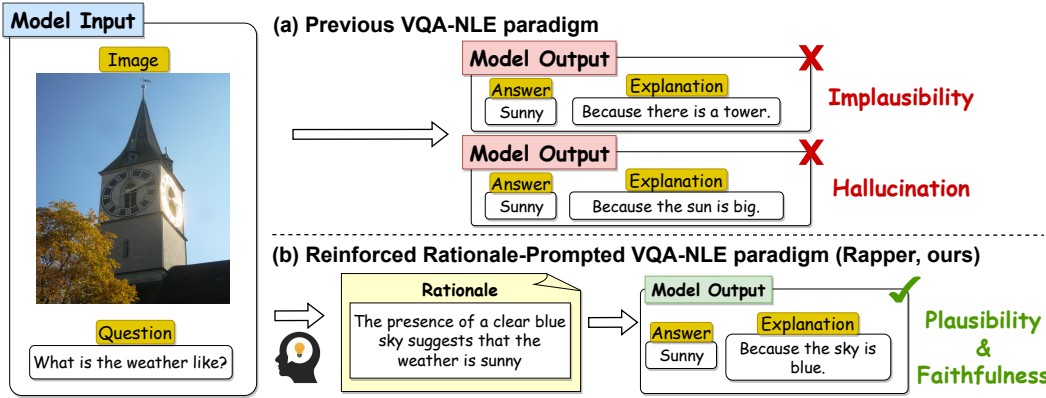

Figure 1: Comparison between (a) previous VQA-NLE paradigm and (b) our proposed reinforced rationale-prompted VQA-NLE paradigm of (*Rapper*). Instead of directly generating answer or explanation, *Rapper* learns plausible and faithful explanations which prompt the VQA model with improved performance.

is that the explanation is not related to the visual image (Ji et al., 2023). More specifically, the former problem is referred to as *implausibility*, while the latter is known as *hallucination*. Take visual input and question in Fig. 1 as an example, "Because there is a tower.' is an *implausible* explanation since it is irrelevant to question, and "Because the sun is big." is a *hallucinated* one since the sun is not visible in the image. Although these issues have been recently studied in the NLE community (Zhao et al., 2023; Turpin et al., 2023), they remain unexplored in the field of VL-NLE. As a result, generating *plausible* yet *faithful* NLEs for elucidating vision-language models continues to pose a crucial challenge.

Recently, rationale-based prompting techniques have been manifested to improve the capability of Large Language Models (LLMs) on complex reasoning tasks (Wei et al., 2022; Liu et al., 2022b). Such techniques involve elicitation of rationales from LLMs, producing knowledge-riched or fact-based intermediate to facilitate the reasoning capability of language model. Thus, these prompting manners are emerging as promising solutions for NLE (Zhao et al., 2023; Krishna et al., 2023). These rationale-prompting paradigms have been further extended to multi-modal regimes such as mm-CoT (Zhang et al., 2023) and mm-ReAct (Yang et al., 2023). However, mm-CoT (Zhang et al., 2023) relies on the ground-truth rationales for training, while mm-ReAct (Yang et al., 2023) have potential hallucinated outputs due to the information loss when converting visual signals into text for ChatGPT API call understanding.

In this paper, we propose **R**einforced **R**ationale-**P**rompted **P**ara**r**digm (*Rapper*) for providing accurate answers for VQA with sufficient NLE, which are plausible and faithful. As depicted in Fig. 1(b), our *Rapper* learns to exploit knowledge learned from LLM and incorporate the corresponding visual content from input images into *rationales* through two stages. Without observing any ground truth rationale during training, the first stage utilizes a knowledge distillation process to introduce LLM for enriching the rationales with supporting facts, encouraging NLE to be factual and plausible. The subsequent stage of *Reinforcement Learning from NLE Feedback* (RLNF) further exploits the answer-explanation feedback to enforce the produced rationales associated with both question and visual inputs, allowing faithful NLE.

We now summarize the contributions of this work below:

- A reinforced rationale-prompted paradigm, *Rapper*, is proposed for plausible and faithful NLE generation in VQA. This is achieved through two proposed stages: knowledge distillation process from LLM and *Reinforcement Learning from NLE Feedback* (*RLNF*).

- In *Rapper*, we first advance LLM and perform knowledge distillation. This results in predicted rationales being based on language-based facts, which prompt the VQA model for plausible NLE.

- To align NLE with the visual input, we introduce *Reinforcement Learning from NLE Feedback* (RLNF) to *Rapper*, which utilizes the answer-explanation feedback as rewards and prompts the VQA model with predicted rationales for faithful NLE.

- Our *Rapper* achieves new state-of-the-art performance for both VQA-X (Park et al., 2018) and e-SNLI-VE (Kayser et al., 2021) on NLE generation. We also demonstrate that *Rapper* outperforms existing VQA-NLE works with reduced implausibility and hallucination.

## 2  RELATED WORK

**Plausible and Faithful Natural Language Explanation**  Research on plausibility and faithfulness in NLE (Majumder et al., 2021; King et al., 2022; Gou et al., 2023; Stacey et al., 2023) has garnered wide attention, particularly due to the evolution of Large Language Models (LLMs) and chain-of-thought (CoT) prompting techniques (Wei et al., 2022). Notably, the method of integrating external knowledge databases for fact generation or retrieval has been proven effective in enhancing the plausibility and faithfulness of NLEs (Majumder et al., 2021; Stacey et al., 2023). Based on this advancement, some recent approaches, such as the *verify-then-correct* pipeline by Gou et al. (2023) and novel decoding strategies proposed by Lan et al. (2023) and King et al. (2022), aim to mitigate hallucination in textual outputs. However, these works typically focus on isolated single text modality or rely on static external knowledge databases, limiting its scalability to multimodal data.

**Natural Language Explanation for Vision-Language Tasks**  Most existing VL-NLE works (Wu & Mooney, 2018a; Park et al., 2018; Marasović et al., 2020; Kayser et al., 2021) generate explanations in a predict-then-explain fashion. Specifically, an answer is first predicted by a pre-trained VL model (e.g., UNITER (Chen et al., 2020) or Oscar (Li et al., 2020)), followed by the generation of the corresponding explanation via a separate language decoder (e.g., GPT2 (Radford et al., 2019)). As the answer and explanation are predicted separately, the explanation often contains irrelevant or contradictory descriptions of the given visual information, struggling to faithfully represent the underlying reasoning process. Recently, NLX-GPT (Sammani et al., 2022) proposes to jointly generate the answer and explanation by a unified sequence-to-sequence model, while S3C (Suo et al., 2023) further enforces the explanation to be consistent with the predicted answer. Although the above approaches have been shown to mitigate the hallucination issue, it is not clear how their NLE is established upon supporting facts or taking the visual input into consideration. Therefore, how to tackle the potential implausibile or hallucinated NLE remains a challenging task.

**Reinforcement Learning for Language Models**  Several research works have explored RL and view it as the key component to enhance models across vision-language tasks such as image captioning (Rennie et al., 2017), novel object captioning (NOC) (Yang et al., 2022), and VQA (Lu et al., 2022a; Fan et al., 2018; Liu et al., 2018). There has been a concentrated effort to align LMs with natural language (NL) feedback (Akyürek et al., 2023; Yang et al., 2022; Liu et al., 2022a) as well as non-NL feedback (Bai et al., 2022; Lu et al., 2022b). For example, Liu et al. (2022a) utilizes the probability of the correct answer as a reward to stimulate an auxiliary module to produce beneficial knowledge, thereby enhancing QA-task performance. Similarly, Yang et al. (2022) employs a CIDEr optimization strategy to enhance the caption with sufficiently visual fidelity in the task of novel object captioning. Despite of their effectiveness, their RL framework or NL-feedback approaches cannot be easily applied for VL-NLE tasks.

## 3  PROPOSED METHOD

### 3.1  PROBLEM FORMULATION

Given a VQA input $X = (V, Q)$ consisting of an input image $V$ and a textual input $Q$ (i.e., question), our goal is to predict the answer $\hat{A}$ and the corresponding explanation $\hat{E}$ (denoted as $\hat{Y} = (\hat{A}, \hat{E})$) via a reasoning module $M$. In order to encourage $M$ to provide correct answer with plausible and faithful explanation, we propose a **Reinforced Rationale-Prompted Paradigm** (*Rapper*) scheme, which learns an additional rationale generator $G$ to jointly exploit the supporting facts from LLMs and the visual content observed from the conditioned image into rationales. Note that only the ground truth $A$ and $E$ are available during training, not the rationales. As depicted in Fig. 2, the learning of *Rapper* is decomposed into: (A) *Knowledge Distillation from LLM* (Sec. 3.2), and (B) *Reinforcement learning from NLE Feedback (RLNF)* (Sec. 3.3), which trains rationale generator $G$ for providing auxiliary intermediates when predicting $\hat{Y} = (\hat{A}, \hat{E})$.

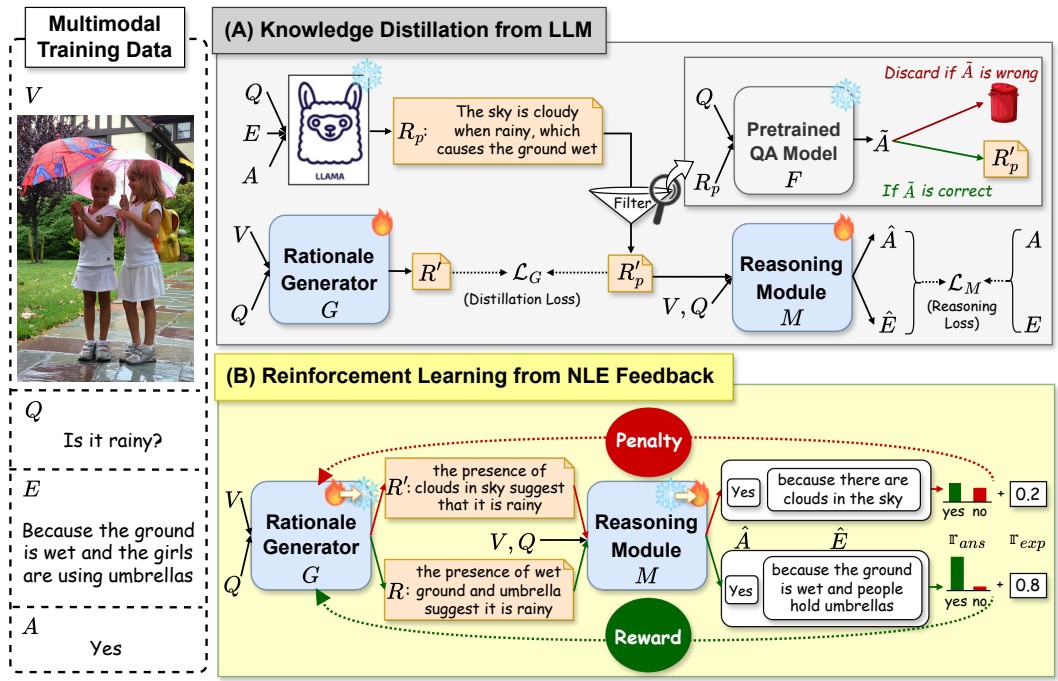

Figure 2: Overview of *Rapper*. *Rapper* involves two training stages: (A) *Knowledge distillation* introduce the rationales $R'_p$ from LLM by offering established facts, facilitating the generation of plausible NLEs from the reasoning module $M$. (B) *Reinforcement learning from NLE feedback* (*RLNF*) further refines the rationales from $R'$ to $R$ by incorporating visual information, encouraging generation of faithful NLEs from $M$.

## 3.2 PLAUSIBLE NLE GENERATION

Since VQA-NLE models typically rely on ground truth answers and explanations for training, it is not clear whether the underlying visual and language knowledge are exploited to support the predicted outputs. In the first stage of *Rapper*, we propose to leverage powerful *reasoning* capability inherent in LLM for plausible NLE generation. As depicted in Fig. 2(A), we propose to learn a rationale generator $G$ by utilizing knowledge distillation from LLM (e.g., LLaMA-65B (Touvron et al., 2023)). This would have the reasoning module $M$ elaborate the conditioned rationales before answering and explaining and encourage plausible NLE. We now detail this learning stage.

### 3.2.1 KNOWLEDGE DISTILLATION FOR FACTED-BASED RATIONALE GENERATION

With the recent success of LLMs showing great capability for generating rationale prompts as intermediate reasoning steps and knowledge (Wei et al., 2022; Kojima et al., 2022; Liu et al., 2022b) for reasoning task, we propose to advance the guidance of pre-trained LLMs to acquire such knowledge, so that supporting facts or knowledge can be exploited and serve as rationales for VL-NLE. Since no ground-truth rationales are available, we leverage the LLM to produce rationales as pseudo ground truth for training our rationale generator $G$. Inspired by Liu et al. (2022a;b) and Min et al. (2022), we elicit pseudo rationale $r_p$ from LLM with a task-specific set of few-shot demonstrations (see Sec. A.5 for details) as follows:

$$R_p = \{r_p \mid r_p \sim P_{LLM}(y, q)\}, \tag{1}$$

where $y$ is the ground-truth answer-explanation pair, $q$ is question, $P_{LLM}$ denotes the LLM in an autoregressive manner, $r_p$ is the sampled pseudo rationale from $P_{LLM}$, and thus $R_p$ is the set of all $r_p$.

However, the above pseudo rationales may be redundant, noisy or lengthy, which would not be desirable for subsequent NLE tasks (Li et al., 2023b). Thus, we apply a post-processing mechanism to filter pseudo rationales $R_p$ to $R'_p$. To be specific, we apply a round-trip consistency by answering

the input question on the pseudo rationales with a pre-trained question-answering (QA) model $F$ [1]. The pseudo rationale is retained when the matching score between the ground-truth answer and the answer predicted by $F$ exceeds a predetermined threshold $\tau$. This matching score is quantified with the token-level F1 score (Wang et al., 2020). Thus, the process of collecting the filtered pseudo rationales $R'_p$ is formulated as follows:

$$R'_p = \{r_p \mid \texttt{F1-score}(\tilde{a}, a) \geq \tau, \; \tilde{a} \sim P_F(Q, r_p), r_p \in R_p\}, \tag{2}$$

where $a$ is the ground truth answer, $\tilde{a}$ is the answer predicted by $F$ based on the pseudo rationale, and $P_F$ denotes the pre-trained QA model $F$ in an autoregressive fashion.

With the above $R'_p$ serving as psuedo ground truth, we are able to train the rationale generator $G$ with the distillation loss $\mathcal{L}_G$ described below:

$$\mathcal{L}_G = -\sum_{t=1}^{T} \log(p_G(r'_{p,t}|r'_{p,0:t-1}, x)), \tag{3}$$

where $r'_p \in R'_p$, $T = |r'_p|$, and $x = \{v, q\} \in X$.

### 3.2.2 PROMPTING BY FACT-BASED RATIONALE FOR PLAUSIBLE NLE

With rationales $R'_p$ better aligned with the facts, we can proceed to the training of the reasoning module $M$ for NLE generation. We note that, since rationales $R'_p$ are in the form of natural language, our the reasoning module $M$ (which is also based on visual-language model) would be able to interpret them. Thus, in addition to the image-question pair $X$ as the inputs to the reasoning module $M$, the derived pseudo rationales $R'_p$ are further viewed as input prompts, which provide fact-supporting conditions when training $M$ to perform VQA-NLE. As a result, we train $M$ by calculating the reasoning loss $L_M$ as follows:

$$\mathcal{L}_M = -\sum_{t=1}^{T} \log(p_M(y_t|y_{0:t-1}, r'_p, x)). \tag{4}$$

In the above cross-entropy loss, $y = [a; e] \in Y$ is the concatenation of the ground-truth answer $a$ and explanation $e$.

### 3.3 FAITHFUL NLE GENERATION

Although the above knowledge distillation process based on LLM introduces plausibility into our rationale generation, the predicted rationales might not be related to the visual input and thus encounter the hallucination problem. To tackle this issue, we introduce a novel technique of *Reinforcement Learning from NLE Feedback* (RLNF). This learning strategy is to encourage the rationale generator $G$ to fully exploit multimodal input data, so that the output rationales are not only plausible but also faithful. Once $G$ produces faithful rationales, we can fine-tune the reasoning module $M$ for plausible yet faithful NLE.

### 3.3.1 RLNF FOR INJECTING VISUAL FACTS

To address the potential hallucination issue, we propose *Reinforcement Learning from NLE Feedback* (*RLNF*) by enforcing rationale generator $G$ to derive the visual facts from the input image into rationales. To achieve this, we define a reward function via RL that penalizes the fact-based but hallucinated rationales $R'$, while rewarding the rationales $R$ that contain both established facts and visual content, as depicted in Fig. 2(B). To achieve this, we design our reward $r_{total}$ to be the addition of answer scores $r_{ans}$ and the explanation score $r_{exp}$, which are the average predicted probability of the ground-truth answer and CIDEr score (Vedantam et al., 2015), respectively. For the answer score, inspired by and following Kadavath et al. (2022), we maximize the answer score to assess the faithfulness of the predicted explanation. This maximization enforces the rationale

---

[1] In the implementation, we follow (Changpinyo et al., 2022) and use UnifiedQA (Khashabi et al., 2022) as the pre-trained QA model.

---

**Algorithm 1** Training RAPPER

---

**Input:** Rationale generator $G$, reasoning module $M$, LLM $P_{LLM}$ and pre-trained QA model $P_F$
**Data:** Image-question pairs $X = \{x^i\}_{i=1}^N$, and answer-explanation pairs $Y = \{y^i\}_{i=1}^N$

   */* Stage(A): KD for Plausible NLE Generation */*
   $R_p \leftarrow$ Collect pseudo rationales (Eq. equation 1);
   $R_p' \leftarrow$ Get filtered pseudo rationales from $R_p$ (Eq. equation 2);
                                                    ▷ Section 3.2.1

   $G \leftarrow$ Update $G$ with $\mathcal{L}_G$ (Eq. equation 3);
   $M \leftarrow$ Update $M$ with $\mathcal{L}_M$ (Eq. equation 4);
                                                    ▷ Section 3.2.2

   */* Stage(B): RLNF for Faithful NLE Generation */*
   $G \leftarrow$ Update $G$ with $\mathbb{R}_{total}$ (Eq. equation 8);            ▷ Section 3.3.1
   $M \leftarrow$ Update $M$ with $\mathcal{L}_M$ (Eq. equation 10);        ▷ Section 3.3.2

**Output**: $G_\theta$, $M_\phi$

---

generator $G$ to inject more visual content into the rationale because the reasoning module $M$ need more visual clues to correctly answer the question. Therefore, this process transform $R'$ to $R$, and simultaneously provide the $M$ with more visual fact-based rationale $R$ to enable the explanation with sufficient faithfulness. On the other hand, the explanation score $\mathbb{r}_{exp}$ is (i.e., specifically CIDEr score) to maintain the plausibility of NLE after the first training stage. As a result, the reward $\mathbb{r}_{total}$ is formulated as follows:

$$\mathbb{r}_{total}(x, a, e, \hat{e}, r) = \mathbb{r}_{ans}(a, x, r) + \mathbb{r}_{exp}(e, \hat{e}), \tag{5}$$

$$\mathbb{r}_{ans}(a, x, r) = \mathcal{Z}(P_{M_\phi}(a \mid x, r)), \tag{6}$$

$$\mathbb{r}_{exp}(e, \hat{e}) = \mathcal{Z}(\texttt{CIDEr}(e, \hat{e})), \tag{7}$$

where $x = \{v, q\}$ is the input image-question pair, $a$ denotes the ground-truth answer, $e$ denotes the ground-truth explanation, $\hat{e}$ is the predicted explanation from $M$, and $r \in R$ is the sampled rationales from $G$. Notably, $\mathcal{Z}$ is an input-specific normalization function that follows Deng et al. (2022) to normalize reward for stabilizing the RL training process.

**RLNF Formulation** Our RLNF employs Proximal Policy Optimization (PPO) (Schulman et al., 2017) as the RL algorithm. As the policy model updated, the rationale generator $G$ is to maximize the following reward $\mathbb{R}_{total}$:

$$\max\{\mathbb{R}_{total}(x, a, e, \hat{e}, r)\}, r \sim \prod_{t=1}^{T} P_G(w_t | w_{<t}), \tag{8}$$

where $r = \{w_i\}_{i=0}^T$, $T = |r|$, and $x = \{v, q\}$. However, we need to ensure the generated rationales are understandable by humans and do not deviate too far from the distilled knowledge. To achieve this, we add a KL penalty term between the learned policy $\theta$ and the initial policy $\theta_{init}$ after the knowledge distillation phase. Therefore, the overall reward is defined as:

$$\mathbb{R}_{total}(x, a, e, \hat{e}, r) = \mathbb{r}_{total}(x, a, e, \hat{e}, r) - \alpha \log \frac{p_G(r|x; \theta)}{p_G(r|x; \theta_{init})}, \tag{9}$$

where $\mathbb{R}_{total}(x, a, e, \hat{e}, r)$ is the reward in Eq. 5.

### 3.3.2 PROMPTING BY VISUAL-FACT-BASED RATIONALE FOR FAITHFUL NLE

Once the rationale generator $G$ is trained with the introduced RLNF, it is encouraged to produce visual fact-based rationales $R$ that are encapsulated with established facts and visual content from visual input. Again, since $R$ are natural language prompts, they are inherently interpretable by our reasoning module $M$. Therefore, for the given image-question pairs $X$, we utilize $R$ as part of input prompts during the reasoning process of $M$. This ensures the NLEs from $M$ retain plausibility because of the established supporting facts lies in $R$, together with the enhanced faithfulness because

of the derived visual content embedded in $R$. We optimize $M$ to achieve this with the reasoning loss $\mathcal{L}_M$ defined as follows:

$$\mathcal{L}_M = -\sum_{t=1}^{T} \log(p_M(y_t|y_{0:t-1}, r, x)), \tag{10}$$

where $r \in R$, $x = \{v, q\} \in X$, and $y = [a; e] \in Y$, which is the concatenated ground-truth answer $a$ and explanation $e$ sequence.

Therefore, through the complete *Rapper* training process as outlined in Algorithm 1, VQA-NLE tasks would be successfully enabled with adequate plausibility and faithfulness.

## 3.4 INFERENCE

At inference time, for a given input image-question pair $x \in X$, we first generate rationale $r$ on the fly from the rationale generator $G$:

$$r = \{w_i \mid w_i \sim P_G(w_{<i}, x); i = 0, \ldots, n\},$$

where $r = \{w_i\}_{i=0}^{n}$ is the sampled rationale, $n = |r|$, and $x = \{v, q\}$. Subsequently, we prompt the reasoning module $M$ by concatenating the predicted rationale $\hat{r}$ with the image-question pair $x$ for outputting the final answer and explanation sequence $\hat{y}$. This can be formulated as:

$$\hat{y} = [\hat{a}; \hat{e}] = \{z_i \mid z_i \sim P_M(z_{<i} \mid x, r); i = 0, \ldots, m\},$$

where $m = |\hat{y}|$, and $\hat{y} = \{z_i\}_{i=0}^{m}$ is the concatenated answer and explanation, denoted as $[\hat{a}; \hat{e}]$.

# 4 EXPERIMENTS

## 4.1 DATASET AND SETUP

We follow (Kayser et al., 2021; Sammani et al., 2022; Suo et al., 2023) and consider two VL-NLE datasets. VQA-X (Park et al., 2018) builds upon VQAv2 dataset (Goyal et al., 2017). It is composed of 32.3K samples, divided into 29K for training, 1.4K for validation, and 1.9K for testing. e-SNLI-VE (Kayser et al., 2021) builds upon e-SNLI dataset (Camburu et al., 2018), consisting of 43K image-hypothesis pairs, divided into 40K for training, 1.4K for validation, and 1.6K for testing.

*Rapper* is consists of a rationale generator $G$ and a reasoning module $M$, are both initialized from the pretrained image captioning model (Li et al., 2023a). The LLM for knowledge distillation during stage(A) is LLaMA-65B (Touvron et al., 2023). More implementation details are shown in Sec. A.1.

## 4.2 EVALUATION METRICS

For NLE evaluation, we use BLEU@N (Papineni et al., 2002), METEOR (Banerjee & Lavie, 2005), ROUGE-L (Lin, 2004), CIDEr (Vedantam et al., 2015), and SPICE (Anderson et al., 2016) as the metrics, while using VQA accuracy to evaluate predicted answers. To evaluate the degree of plausibility and faithfulness of explanations, we measure them with CIDEr/SPICE and RefCLIPScore Hessel et al. (2021), respectively. In addition, we build human evaluation for explanation on plausibility and faithfulness since automatic metric measures not always reflect the correctness and logicality. Please refer to Appendix A.3 for the details of our human evaluation process.

**Plausibility**   To quantitatively evaluate explanation plausibility, we employ CIDEr and SPICE scores. CIDEr measures the similarity between the generated explanation and human-written ground truth sentences, capturing human consensus by introducing tf-idf weight (Vedantam et al., 2015). On the other hand, SPICE converts sentences into semantic scene graphs, allowing evaluation to break grammatical constraints and thus closely resembling human judgment (Anderson et al., 2016).

**Faithfulness**   We adopt RefCLIPScore, which computes the harmonic mean of CLIPScore (Hessel et al., 2021) and maximal reference cosine similarity, thereby encapsulating the correlation between the explanation and its reference. As noted by Hessel et al. (2021), RefCLIPScore surpasses prior metrics in correlating with human judgment for hallucination detection.

| Method | VQA-X | | | | | | | | |
|---|---|---|---|---|---|---|---|---|---|
| | B@1 | B@2 | B@3 | B@4 | METEOR | ROUGE-L | CIDEr | SPICE | Accuracy |
| PJ-X (Park et al., 2018) | 57.4 | 42.4 | 30.9 | 22.7 | 19.7 | 46.0 | 82.7 | 17.1 | 76.4 |
| FME (Wu & Mooney, 2018b) | 59.1 | 43.4 | 31.7 | 23.1 | 20.4 | 47.1 | 87.0 | 18.4 | 75.5 |
| RVT (Marasović et al., 2020) | 51.9 | 37.0 | 25.6 | 17.4 | 19.2 | 42.1 | 52.5 | 15.8 | 68.6 |
| QA-only (Kayser et al., 2021) | 51.0 | 36.4 | 25.3 | 17.3 | 18.6 | 41.9 | 49.9 | 14.9 | - |
| e-UG (Kayser et al., 2021) | 57.3 | 42.7 | 31.4 | 23.2 | 22.1 | 45.7 | 74.1 | 20.1 | 80.5 |
| NLX-GPT (Sammani et al., 2022) | 64.2 | 49.5 | 37.6 | 28.5 | 23.1 | 51.5 | 110.6 | 22.1 | 83.07 |
| S3C (Suo et al., 2023) | 64.7 | 50.5 | 38.8 | 30.7 | 23.9 | 52.1 | 116.7 | 23.0 | 85.6 |
| *Rapper* (ours) | **65.5** | **51.6** | **40.5** | **31.8** | **24.3** | **52.9** | **124.0** | **24.5** | **87.25** |

| Method | e-SNLI-VE | | | | | | | | |
|---|---|---|---|---|---|---|---|---|---|
| | B@1 | B@2 | B@3 | B@4 | METEOR | ROUGE-L | CIDEr | SPICE | Accuracy |
| PJ-X (Park et al., 2018) | 29.4 | 18.0 | 11.3 | 7.3 | 14.7 | 28.6 | 72.5 | 24.3 | 69.2 |
| FME (Wu & Mooney, 2018b) | 30.6 | 19.2 | 12.4 | 8.2 | 15.6 | 29.9 | 83.6 | 26.9 | 73.7 |
| RVT (Marasović et al., 2020) | 29.9 | 19.8 | 13.6 | 9.6 | 18.8 | 27.3 | 81.7 | 32.5 | 72.0 |
| QA-only (Kayser et al., 2021) | 29.8 | 19.7 | 13.5 | 9.5 | 18.7 | 27.0 | 80.4 | 32.1 | - |
| e-UG (Kayser et al., 2021) | 30.1 | 19.9 | 13.7 | 9.6 | 19.6 | 27.8 | 85.9 | 34.5 | **79.5** |
| NLX-GPT (Sammani et al., 2022) | 37.0 | 25.3 | 17.9 | 12.9 | 18.8 | 34.2 | 117.4 | 33.6 | 73.91 |
| *Rapper* (ours) | **40.5** | **28.1** | **20.2** | **14.7** | **20.8** | **35.9** | **128.6** | **34.9** | 75.73 |

Table 1: Quantitative NLE comparisons of *filtered* results (i.e., NLE evaluation conditioned on correct answers) on VQA-X and e-SNLI-VE.

| Method | Unfiltered | | | | | Filtered | | | | | Accuracy |
|---|---|---|---|---|---|---|---|---|---|---|---|
| | B@4 | METEOR | ROUGE-L | CIDEr | SPICE | B@4 | METEOR | ROUGE-L | CIDEr | SPICE | |
| *Rapper* | **30.0** | 23.3 | **51.3** | **116.0** | 23.2 | **31.8** | 24.3 | **52.9** | **124.0** | 24.5 | **87.3** |
| − RLNF | 29.4 | **23.6** | 51.2 | 113.0 | 23.0 | 31.2 | **24.5** | 52.5 | 120.2 | 24.2 | 86.6 |
| − RLNF − KD | 27.1 | 21.8 | 49.7 | 103.2 | 20.7 | 29.3 | 23.0 | 51.6 | 112.1 | 22.3 | 85.0 |

| Method | Unfiltered | | | | | Filtered | | | | | Accuracy |
|---|---|---|---|---|---|---|---|---|---|---|---|
| | B@4 | METEOR | ROUGE-L | CIDEr | SPICE | B@4 | METEOR | ROUGE-L | CIDEr | SPICE | |
| *Rapper* | **30.0** | **23.3** | **51.3** | **116.0** | **23.2** | **31.8** | **24.3** | **52.9** | **124.0** | **24.5** | **87.3** |
| *Rapper* w/o filtering | 28.5 | 22.7 | 50.8 | 110.6 | 22.2 | 30.1 | 23.4 | 52.1 | 116.7 | 23.4 | 86.4 |

Table 2: Ablation studies of the proposed training schemes (up) and the filtering mechanism for knowledge distillation (bottom). We compare the performances in both filtered and unfiltered settings.

## 4.3 QUANTITATIVE ANALYSIS

**NLE evaluation.** In Table 1, Table 5, and Table 6, we demonstrate that *Rapper* outperform previous state-of-the-art methods in NLE-related metrics on both VQA-X and e-SNLIV-VE datasets, with *filtered* and *unfiltered* settings. The *filtered* setting in Table 1 considers the explanations that are associated with correct answers. Conversely, the *unfiltered* setting in Table 5 and Table 6 in Appendix A.2 indicates evaluations of explanations without considering the correctness of the corresponding answers.

| Method | RefCLIPScore(↑) |
|---|---|
| *Much recent VL-NLE works* | |
| NLX-GPT | 64.06 |
| S3C | 65.09 |
| *Our stage-ablated approaches* | |
| *Rapper* (w/o KD and w/o RLNF) | 66.00 |
| *Rapper* (w/o RLNF) | 65.66 |
| *Rapper* | **67.05** |

Table 3: Faithfulness evaluation on the VQA-X dataset under filtered setting. Note that a higher RefCLIPScore indicates less hallucination.

**Plausibility & faithfulness of NLE.** We assess the plausibility and faithfulness in NLE through CIDEr/SPICE (in Table 1), RefCLIPScore (in Table 3), and human evaluation (in Table 4). In table 1, we demonstrate that *Rapper* outperforms previous state-of-the-art methods in NLG metrics on both VQA-X and e-SNLI-VE benchmarks, underscoring its superiority in generating plausible explanations.

| Method | Plausibility (↑) | Faithfulness (↑) |
|---|---|---|
| NLX-GPT | 0.771 | 0.795 |
| S3C | 0.797 | 0.811 |
| *Rapper* | **0.845** | **0.859** |

Table 4: Human evaluation on plausibility and faithfulness on VQA-X in the filtered setting.

On the other hand, in table 3, *Rapper*'s superior RefCLIPScore indicates fewer hallucinations and increased faithfulness over other VQA-NLE works, although the RefCLIPScore of *Rapper* (w/o RLNF) is lower due to the hallucinations introduced by knowledge distillation from LLM. Nonetheless, *Rapper* still successfully reduce hallucination after the RLNF training. This demonstrates the effectiveness of our proposed RLNF to enable the model to generate faithful NLEs. Lastly, through human evaluation in Table 4, we provide further human-perceived evidence for the effectiveness of *Rapper* for improved NLE generation.

| Multimodal Input / Methods | | (a) | (b) | (c) |
|---|---|---|---|---|
| | | **Q**: Is the table cluttered?
**GT A**: No
**GT E**: There is only a single vase with flowers on it | **Q**: Is this in an asian country?
**GT A**: Yes
**GT E**: there is an asian language used as text font in public | **Q**: What kind of animal is this?
**GT A**: Sheep
**GT E**: The animal is covered in thick wool |
| NLX-GPT | $\hat{A}$ | No | Yes | Sheep |
| | $\hat{E}$ | There are no objects in the table | There is a train on the tracks | It has a long face and long nose |
| S³C | $\hat{A}$ | No | Yes | Sheep |
| | $\hat{E}$ | There are only a few items on it | There is a train in the stations | It has a long snout and white fur |
| Rapper | $R$ | The table is not cluttered because there is only one object on it | The presence of asian writing on the train suggests that it is in an asian country | A sheep is a type of animal that has wool on its body |
| | $\hat{A}$ | No | Yes | Sheep |
| | $\hat{E}$ | There is only one object on it | There is asian writing on the train | Its has wool on its body |

Figure 3: Visualization of output answers and explanations predicted by different methods. Note that words in red denote hallucinated explanations, and those in orange denote implausible ones. Words in blue denote faithful and plausible explanations to the input image-question pair.

**Ablation on the proposed stages.**  In top of Table 2, we evaluate our two-stage approach: (A) KD from LLM and (B) RLNF. Compared to the *Rapper* baseline without KD and RLNF, our method enhances explanation plausibility and faithfulness, highlighting the importance of both stages.

**Ablation on the "Filter" mechanism.**  In bottom of Table 2, our filtering mechanism in knowledge distillation stage outperforms the baseline *Rapper* without filtering, by effectively removing overly redundant and noisy pseudo rationales that could impair model performance.

**Ablation studies of derived rationales.**  In Fig. 4, we demonstrate that introducing two proposed stages improves the quality of derived rationales, benefiting the VQA performance of vision-language large model. Specifically, we test whether mPLUG-Owl (Ye et al., 2023) can answer accurately when given a pair (image, question, and $x \in (None, R', R)$), where $x = None$ indicates no rationales as input, $x = R'$ indicates the rationales are from *Rapper* with KD training, and $x = R$ indicates the rationales are from *Rapper* with both KD and RLNF training. Notably, we find that rationale quality improves progressively as we implement the stages we have proposed. This underscores the effectiveness of our designed stages in enhancing rationale quality.

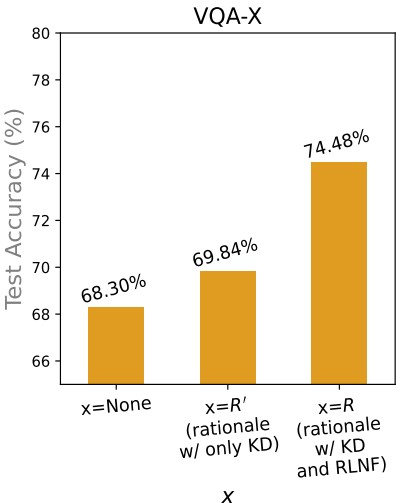

Figure 4: Ablation studies of derived rationales. Note the VQA accuracy on the VQA-X dataset is evaluated.

## 4.4 QUALITATIVE EVALUATION

In Fig.3, we compare NLX-GPT(Sammani et al., 2022), S3C (Suo et al., 2023), and our *Rapper* on the VQA-X dataset. *Rapper* consistently produces more plausible explanations. For example, Fig.3(a) highlights ability of *Rapper* to derive visual facts, such as identifying a single object on the table, surpassing previous methods that might produce hallucinated explanations. Similarly, in Fig.3(b), *Rapper* offers plausible explanations like recognizing Asian writing, contrasting with the implausible outputs of prior methods. Additional results and ablation studies are in Appendix A.4.

## 5 CONCLUSION

In this paper, we proposed *Rapper*, a two-stage Reinforced Rationale-Prompted Paradigm for enabling NLE with sufficient plausible and faithful properties. Our *Rapper* uniquely distills language-based knowledge from LLM and utilizes RL with natural language feedback from the VQA task, so that the designed rationale generator is able to produce rationales with the aforementioned desirable properties. By prompting such predicted rationales into the reasoning module, we demonstrated that satisfactory VQA performances can be achieved. Compared to SOTA VQA-NLE methods, possible implausible or hallucinated explanations can be mitigated by our *Rapper*.

ACKNOWLEDGMENTS AND DISCLOSURE OF FUNDING

This work is supported in part by the National Science and Technology Council via grant NSCT112-2634-F-002-007 and Center of Data Intelligence: Technologies, Applications, and Systems via grant NTU-113L900902. We also thank the National Center for High-performance Computing (NCHC) for providing computational and storage resources.

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

# A  APPENDIX

## A.1  IMPLEMENTATION DETAILS

Given an image-question pair $x = (v, q) \in X$, the rationale generator $G$ first generate rationales $r \in R$ that contains both rich knowledge and visul-grounded facts. The ground-truth answer-explanation pair $y = (a, e) \in Y$ is available. Subsequently, conditioned on these rationales as well as the image-question pair, the reasoning module is enable to generate the answer $\hat{a} \in A$ as well as NLEs $\hat{e} \in E$ that are sufficiently plausible and faithful. Specifically, our approach follows a series of steps outlined in Algorithm 1. We use 8 V100 GPUs to perform the above training algorithms.

**Stage(A): KD from LLM**  In Sec. 3.2.1, we gather pseudo rationales for each image-question pairs using in-context learning to prompt LLaMA-65B (Touvron et al., 2023). To ensure the quality of these pseudo rationales, we employ UnifiedQA (Khashabi et al., 2022) for filtering, keeping only those rationales whose predicted answers have a token-level F1 score surpassing a threshold $\tau$ (follow Changpinyo et al. (2022) to set it to 0.54 manually).Next, proceeding to Sec. 3.2.2, we train rationale generator $G$ for 10 epochs using the distillation loss $\mathcal{L}_G$. The input contains a image and a input textual template, formed by concatenating the question with the filtered pseudo rationale, represented as [Question: $\{q\}$ Rationale: $\{r'_p\}$]. The ground-truth label template is $[r'_p]$. Training settings include a total batch size of 128, a learning rate of 3e-5, and a weight decay of 0.95.

Proceeding to Sec. 3.2.2, we train the reasoning module $M$ with the reasoning loss $\mathcal{L}_M$ for 15 epochs. Similar to the rationale generator, the input template includes the concatenated question and pseudo rationale, which is formulated as [Question: $\{q\}$ Rationale: $\{r'_p\}$ Answer: $\{\hat{a}; \hat{e}\}$], and the ground truth label template is $\{a; e\}$.

**Stage(B): RL for NLE feedback**  In Sec. 3.3.1, we apply *RLNF* to continue to train the rationale generator $G$. For the RL experimental settings, we follow von Werra et al. (2020), and use their default PPO hyperparameter setting to train for 10 epochs with a batch size of 128. The rationale generator $G$ and reasoning module $M$ both use greedy search to sample the rationales, answers, and explanations for RL optimization.

Finally, in Sec. 3.3.2, we continue to train the reasoning module $M$ for 10 epochs with the same loss $\mathcal{L}_M$ and similar training parameters. The input contains a image and a input template which involves the concatenated question $q$, our predicted rationale $\hat{r}$, and ground-truth answer $e$ and explanation $e$, which is formulated as: [Question: $\{q\}$ Rationale: $\{\hat{r}\}$ Answer: $\{a; e\}$], and the ground truth label template is $\{a; e\}$. During the training period, the rationale generator $G$ samples the rationales on the fly by beam search decoding with a beam size of 5. During the evaluation period, both $G$ and $M$ generate rationales and answer-explanation pairs by beam search decoding with a beam size of 5.

## A.2  UNFILTERED QUANTATIVE RESULTS

| Method | VQA-X | | | | |
|---|---|---|---|---|---|
| | B@4 | M | R | C | S |
| CAPS (Park et al., 2018) | 5.9 | 12.6 | 26.3 | 35.2 | 11.9 |
| PJ-X (Park et al., 2018) | 19.5 | 18.2 | 43.4 | 71.3 | 15.1 |
| FME (Wu & Mooney, 2018b) | 24.4 | 19.5 | 47.7 | 88.8 | 17.9 |
| NLX-GPT (Sammani et al., 2022) | 25.6 | 21.5 | 48.7 | 97.2 | 20.2 |
| S3C (Suo et al., 2023) | 27.8 | 22.8 | 50.7 | 104.4 | 21.5 |
| *Rapper* (ours) | **30.0** | **23.3** | **51.3** | **116.0** | **23.2** |

| Method | e-SNLI-VE | | | | |
|---|---|---|---|---|---|
| | B@4 | M | R | C | S |
| NLX-GPT (Sammani et al., 2022) | 11.9 | 18.2 | 32.5 | 109.0 | 33.0 |
| *Rapper* (ours) | **13.9** | **20.1** | **34.6** | **121.6** | **34.9** |

Table 5: Quantitative comparisons of *unfiltered* scores on VQA-X dataset.

Table 6: Quantitative comparisons of *unfiltered* scores on e-SNLI-VE dataset.

As demonstrated in Table 5 and Table 6, RAPPER significantly outperforms existing VL-NLE methods across all metrics on both VQA-X and e-SNLI-VE datasets. It's notable that *Rapper* surpasses the second-best results by 11.6 and 12.6 in CIDEr on VQA-X and e-SNLI-VE datasets, representing a relative improvement of 11.1% and 11.6%, respectively.

## A.3 Human Evaluation Process

We follow the evaluation setting/process applied in NLXGPT (Sammani et al., 2022) and S3C (Suo et al., 2023) (i.e., two SOTAs on VQA-NLE), we randomly select 200 test samples from the VQA-X (Park et al., 2018) dataset with correctly predicted answers. Then, subjective evaluation is performed by 3 different annotators. Note that each annotator has to select one out of 4 choices: yes, weak yes, weak no, and no, as a response to whether the explanation justifies the answer. And, these 4 decisions are numerically mapped to 1, 2/3, 1/3, and 0, respectively. With averaged results obtained for each method, we present the performance comparisons in the following table. From this table, we see our proposed Rapper is preferable by the users in terms of subjective plausibility and faithfulness assessment. This conclusion also aligns with the objective quantification evaluation of Table 1 presented in the main paper.

## A.4 More Qualitative Results

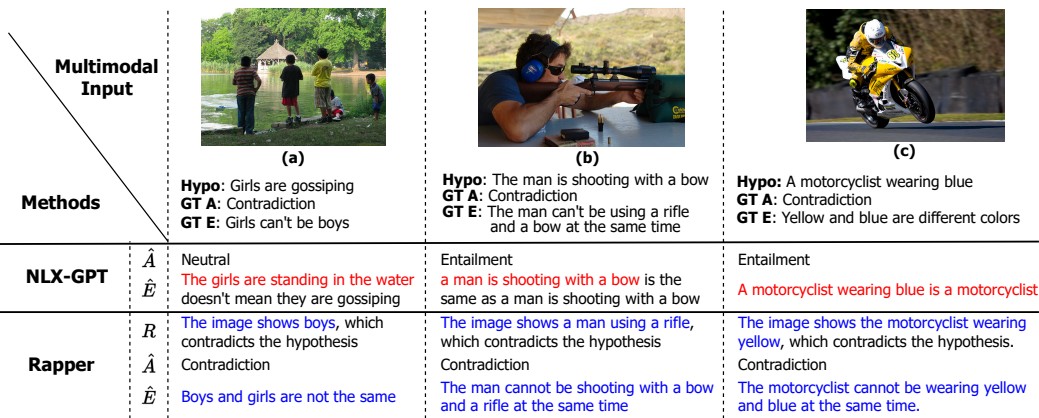

Figure 5: Visualization of output answers and explanations predicted by different methods on e-SNLI-VE dataset. Note that words in red denote hallucinated explanations and those in blue denote faithful and plausible explanations to image.

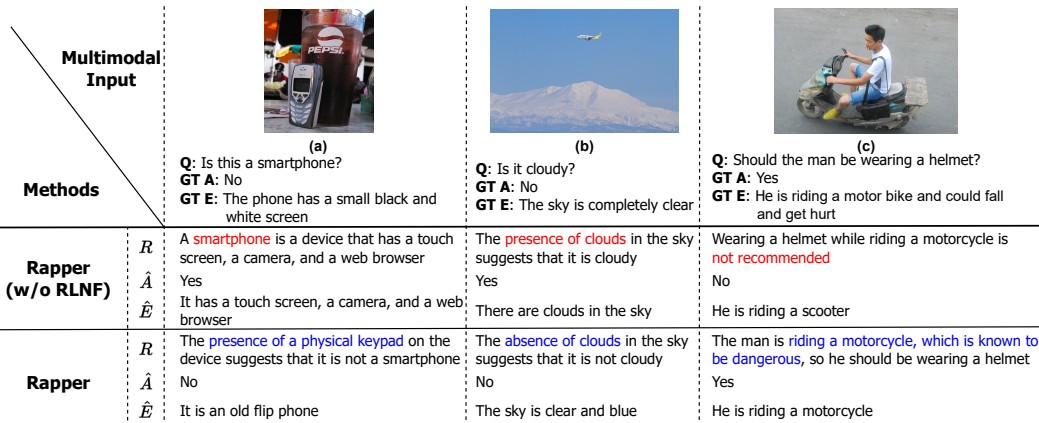

Figure 6: Visualization of output rationale, answer, and explanations predicted by the different training stages (A) and (B). Note that words in red denote hallucinated rationales, while those in blue are rationales that derive visual facts from the input image.

In Fig. 5, we present some qualitative results comparing NLX-GPT (Sammani et al., 2022) and our *Rapper* on the e-SNLI-VE dataset. The results demonstrate that *Rapper* generates more precise

answers accompanied by more faithful explanations. For instance, as seen in Fig. 5(b), NLX-GPT mistakenly identifies the rifle in the image as a bow. On the other hand, *Rapper* accurately answers the question, supported by the visually evident fact contained within the prompting rationale that the man is holding a rifle. As shown in Fig. 6, we present some qualitative results comparing *Rapper* without *RLNF* and our *Rapper* on the VQA-X dataset. Through the comparison, we can observe that *RLNF* help lower our model's probability of generating hallucinated rationales. As a result, our model can generate more accurate answers and reasonable explanations with the help of *RLNF*.

We show more qualitative results in Fig. 7.

| # | Image | Question | Rationale | Answer/Explanaiton | GT Answer/GT Explanation |
|---|-------|----------|-----------|--------------------|--------------------------|
| 1 | | Should the cars be stopped at this light? | a green light indicates that cars are allowed to proceed through the intersection, and a red light indicates that cars are not allowed to proceed through the intersection | Yes / The light is red | Yes / the light is a red color |
| 2 | | What is this man doing? | the presence of a baseball in the pitcher's hand suggests that the man is pitching a baseball | Pitching / He is about to throw a baseball | Pitching / He is throwing a baseball |
| 3 | | What event is happening? | The the presence of graduation caps and gowns suggests that a graduation is taking place | Graduation / People are wearing graduation caps and gowns | Graduation / They are all wearing graduation gowns |
| 4 | | What kind of room is this? | The presence of a toilet in the room suggests that it is a bathroom | Bathroom / There is a toilet in it | Bathroom / There is a toilet in a closet |
| 5 | | Does the guy look happy? | an angry expression suggests that the guy is not happy | No / he has an angry look on his face | No / he isn't smiling and look annoyed |
| 6 | | What is he doing? | he presence of a person cooking suggests that the person is cooking | Cooking / He is using a pan to cook food | Cooking / he is using a pan on the stove |
| 7 | | What sport are they paparticipating in? | The presence of skis and poles in the picture suggests that they are participating in skiing | Skiing / They have skis on their feet and poles in their hands | Skiing / There is a skis under each feet |
| 8 | | What kind of sandwich is this? | The presence of hot dogs in the sandwich suggests that it is a hot dog sandwich | Hot dog / There are hot dogs in it | Hot dog / Their is hot dog mean in a bun |
| 9 | | What sport is being played? | The presence of a tennis racket suggests that the sport being played is tennis | Tennis / The man is holding a tennis racket | Tennis / The player is holding a tennis racket |

Figure 7: More qualitative results of *Rapper* compares to ground truth answers and explanations.

## A.5 FEW-SHOT DEMONSTRATIONS FOR ELICITING LLAMA TO GENERATE PSEUDO RATIONALES

In Fig. 8 and Fig. 9, we show the task-specific few-shot demonstrations for the VQA-X and e-SNLI-VE tasks. We use these demonstrations to prompt LLaMA (Touvron et al., 2023) with in-context learning.

Please gives the rationale that justifies the following explanation and answer, according to the question

question: does the device in the foreground run on water
answer: no
explanation: bicycles can not be ridden on the water
rationale: the rationale is that bicycles are designed to be operated on solid ground, so it is unlikely that it would run on water.

question: does this kid look excited
answer: yes
explanation: the boy has his arms in the air and a smile on his face
rationale: the rationale is that arms in the air and a smile are indications of excitement

question: is this a recent picture
answer: no
explanation: because the tennis player has a modern haircut and is holding a modern racket
rationale: the rationale is that modern hairstyles and equipment are not typically seen in older photos, suggesting that this is not a recent picture

question: should this cake be eaten by only one person
answer: no
explanation: because it is a large cake for a party
rationale: the rationale is that large cakes are typically intended to be shared and enjoyed by multiple people, so they should not be eaten by only one person

Figure 8: Few-shot demonstrations for prompting LLaMA (Touvron et al., 2023) to generate pseudo rationales in VQA-X task.

Please give the rationale according to the hypothesis, answer, and explanation.

hypothesis: A dog and a pig play in some mud
answer: contradiction
explanation: a boy is not a dog
rationale: the rationale is that the dog is not exist in the image, so the two statement is contradict.

hypothesis: A clean shaven man is standing in a cornfield
answer: contradiction
explanation: a man with a goatee is not a clean shaven man
rationale: the rationale is that the clean shaven man doesn't have any facial hair who doesn't have goatee in the image which is contradicted to the hypothesis.

hypothesis: The people are eating in the kitchen
answer: contradiction
explanation: conversing and eating are different actions
rationale: the rationale is that the people are conserving instead of eating in the image, which is contradicted to the hypothesis.

hypothesis: There are men outside
answer: entailment
explanation: men clean the outside windows means there are men outside
rationale: the rationale is that men clean the outside windows, impling they are outside.

hypothesis: Two men are sitting on chairs.
answer: contradiction
explanation: people who are sitting than can not be in a marathon at the same time.
rationale: the rationale is that the image shows people running a marathon and there are no chairs in the image, which implies they are not sitting on chairs.

hypothesis: A person is outside.
answer: entailment
explanation: you have to be outside to be under a tree.
rationale: the rationale is that trees are typically found outdoors, which implies they are outside.

hypothesis: A group of people are walking to work.
answer: neutral
explanation: just because walking down the sidewalk does not mean they are walking to work
rationale: the rationale is that there is no clear indication of people's destination in the image, the hypothesis remains neutral.

hypothesis: The three young men are riding their own private sailboat.
answer: contradiction
explanation: they cannot simultaneously be riding their own private boat and taking public transportation.
rationale: the rationale is that the image shows the three young men taking public transportation, which contradicts the hypothesis.

hypothesis: There are no women near the birds on the steps.
answer: contradiction
explanation: if there is a woman in black then you cannot say no women are there.
rationale: the rationale is that the image shows a woman in black clothing standing near the birds on the steps, which contradicts the hypothesis.

hypothesis: The man stands on the ladder while someone holds it steady.
answer: entailment
explanation: person holding a ladder means that someone is holding the ladder steady
rationale: the rationale is that the image shows someone holding the ladder, which implies that the ladder is being held steady, and therefore supports the hypothesis.

hypothesis: The people are all friends.
answer: neutral
explanation: just because two girls swing on with a boy doesn't imply they are friends.
rationale: the rationale is that there is no clear indication of the relationship between the people in the image, so the hypothesis remains neutral.

hypothesis: Girl plays an instrument.
answer: entailment
explanation: the drums are an instrument girls are children therefore they are young.
rationale: the rationale is that the image shows a girl playing drums, which supports the hypothesis.

hypothesis: A boy with a blue towel and white hat and goggles is getting ready to go home for the day.
answer: contradiction
explanation: the boy holding goggles cannot wear them at the same time.
rationale: the rationale is that the image shows a boy holding goggles on his head, which contradicts the hypothesis.

hypothesis: A boy on a skateboard riding down the street his house is on.
answer: neutral
explanation: it can not be assumed that it is the street his house is on.
rationale: the rationale is that there is no clear indication in the image that the street is where the boy's house is located, so the hypothesis remains neutral.

hypothesis: Workers clean up after a bomb explodes.
answer: neutral
explanation: there may only be the crane operator, not multiple workers the crane could be in the midst of a build, not necessarily a clean up rubble occurs for more reasons than when a bomb explodes
rationale: the rationale is that there is no clear indication of numbers of workers, worker's activity and evidence of bombing in the image.

hypothesis: The male child is on a beach.
answer: entailment
explanation: asian boy with spiky hair, wearing a yellow shirt smilies on the beach.
rationale: the rationale is that the image shows a male child with spiky hair wearing a yellow shirt and smiling on a beach, which supports the hypothesis.

Figure 9: Few-shot demonstrations for prompting LLaMA (Touvron et al., 2023) to generate pseudo rationales in e-SNLI-VE task.

