# OpenReview forum: "RAPPER: Reinforced Rationale-Prompted Paradigm for Natural Language Explanation in Visual Question Answering"
_ICLR.cc/2024/Conference — ICLR 2024 poster_

### Official Review · Reviewer_iXW4 · 2023-10-29

**Soundness:** 3 good
**Presentation:** 2 fair
**Contribution:** 3 good
**Rating:** 6
**Confidence:** 2

**Summary:**

The paper introduces RAPPER, a two-stage Reinforced Rationale-Prompted Paradigm designed to improve Natural Language Explanation (NLE) in Visual Question Answering (VQA) tasks. The first stage employs knowledge distillation from large language models (LLMs) to generate rationales that are fact-based. The second stage introduces a unique Reinforcement Learning from NLE Feedback (RLNF) to incorporate visual facts into the NLE generation. The paper claims that RAPPER outperforms existing state-of-the-art methods in VQA-NLE on two benchmarks, providing more plausible and faithful explanations.

**Strengths:**

The paper addresses the problem of implausibility and hallucination in NLE for VQA, which is a novel contribution. The two-stage approach combining knowledge distillation and reinforcement learning is also unique and the RLNF technique for incorporating visual facts into NLE is particularly noteworthy.

The paper is well-organized and clearly articulates the problem, the proposed solution, and its advantages. The use of figures to illustrate the model architecture and the comparison with existing methods is helpful.

Improving the plausibility and faithfulness of NLE in VQA has important implications for real-world applications, such as medical VQA, where interpretability is crucial.

**Weaknesses:**

The two-stage approach, while novel, adds complexity to the model. It would be beneficial to see a discussion on the trade-offs involved, such as computational cost.

The paper focuses on VQA tasks, and it's not clear how's performance of the proposed method when it was adapted to other vision-language tasks.

No human evaluation is conducted regarding the generation quality.

**Questions:**

How does the complexity of the two-stage approach impact the computational efficiency of the model?

Could you elaborate on how the RLNF stage specifically tackles the hallucination problem in NLE?

---

> ### Author Response · Authors · 2023-11-23
> **Responses to Reviewer iXW4 (1/2)**
>
> **Q1: The two-stage approach, while novel, adds complexity to the model. It would be beneficial to see a discussion on the trade-offs involved, such as computational cost. How does the complexity of the two-stage approach impact the computational efficiency of the model?**
>
> **A1:** We thank the reviewer for suggesting further analysis for the computational cost. To discuss the trade-off between computational cost and our model performance, we now present the total and trainable model parameters of Rapper and its ablated variations (as listed in the upper part of Table 3).
>
> From the table below, we see that Rapper significantly improves the metrics of natural language generation (e.g., CIDEr, SPICE, and Rough-L) over both the Baseline (Rapper w/o KD and w/o RLNF, a one-stage approach) and Rapper w/o RLNF (a two-stage approach). It is worth noting that, the Baseline only deploys the proposed reasoning module $M$ that is based on BLIP2 [1] and has a total of 3.82B parameters, including 187M trainable parameters (i.e., Qformer, initialized from BERT [2]) and 3.63B freezed parameters (i.e., ViT [3] and OPT [4]). On the other hand, Rapper w/o RLNF and the full version of Rapper are composed of the proposed rationale generator $G$ and reasoning module $M$. For such 2-stage frameworks, both modules are based on BLIP2 and share 3.63B freezed parameters, with 187M trainable parameters for each (and thus a total of 374M trainable parameters). It can be seen that, the two-stage methods do require a larger number (2X) of trainable parameters than the one-stage method does. However, the increased amount is in the same order (187M), and it is remarkably less than the total amount of parameters (~4B).
>
> As for the training/inference stages, the runtime estimates are also listed in the table below, which are performed on 8/1 Nvidia V100 GPU(s) on the VQA-X [5] dataset. From the table below, we see that the two-stage method of Rapper w/o RLNF requires 2X computation time for both training and testing, and the full version of Rapper needs additional 3X computation time due to the RL optimization scheme. Nevertheless, even with RLNF, training can be done in 18hrs without utilizing an advanced GPU environment. And, acceleration of LLM training/inference is still an active research topic in the community. We sincerely thank the reviewer for suggesting the above analysis, which will be added to the revised version for the completeness of our discussions.
>
>
> | Method | Total / Trainable param. | Training time (hr) | Inference time (ms/sample) | B@4  | Rough_L | CIDEr | SPICE | Acc. |
> |-|-|-|-|-|-|-|-|-|
> | Baseline (Rapper w/o KD w/o RLNF) | 3.82B / 187M | 3 | 167.7 | 29.3 | 51.6 | 112.1 | 22.3 | 85.0 |
> | Rapper w/o RLNF | 4.01B / 374M | 6 | 291.3 | 31.2 | 52.5 | 120.2 | 24.2  | 86.6 |
> | *Rapper* | 4.01B / 374M | 18 | 291.3 | **31.8** | **52.9** | **124.0** | **24.5**  | **87.3** |
>
> [1] Li, Junnan, et al. Blip-2: Bootstrapping language-image pre-training with frozen image encoders and large language models. ICML 2023.
>
> [2] Devlin et al. Bert: Pre-training of deep bidirectional transformers for language understanding. NAACL 2019.
>
> [3] Dosovitskiy et al. An image is worth 16x16 words: Transformers for image recognition at scale. ICLR 2021.
>
> [4] Zhang et al. Opt: Open pre-trained transformer language models. arXiv 2022.
>
> [5] Park et al. Multimodal explanations: Justifying decisions and pointing to the evidence. CVPR 2018.
>
> **Q2: The paper focuses on VQA. How's performance when it was adapted to other vision-language tasks?**
>
> **A2:** We thank the reviewer for the suggestion. We are happy to point out that we did apply our Rapper on another VL-NLE (natural lange explanation) task of Visual Entailment (VE), as listed in the lower part of Table 1 of our main paper.
>
> We note that, VE requires one to assign a label determining the relationship between a premise image and a textual hypothesis (i.e., “entailment”, “neutral”, and “contradiction”), while giving an explanation for supporting the aforementioned relationship [6]. And, VE is tyically applied in cross-modal retrieval, etc. applications. In Table 1 of our main paper, we report the performance on the **e-SNLI-VE** benchmark, in which we observed that our Rapper performed favorably against NLXGPT [7] and S3C [8] in terms of different NLG metrics (i.e., CIDEr, SPICE, and ROUGE-L). We will add the definition of VE with the above additional clarification to Sect. 4.1 of our main paper.
>
> [6] Do et al. e-snli-ve: Corrected visual-textual entailment with natural language explanations. CVPRW 2020.
>
> [7] Sammani et al. Nlx-gpt: A model for natural language explanations in vision and vision-language tasks. CVPR 2022.
>
> [8] Suo et al. S3C: Semi-Supervised VQA Natural Language Explanation via Self-Critical Learning. CVPR 2023.

---

> > ### Author Response · Authors · 2023-11-23
> > **Responses to Reviewer iXW4 (2/2)**
> >
> > **Q3: No human evaluation is conducted regarding the generation quality.**
> >
> > **A3:** We thank the reviewer for suggesting additional experiments for subjective evaluation. Following the evaluation setting/process applied in NLXGPT [7] and S3C [8] (i.e., two SOTAs on VQA-NLE), we randomly select 200 test samples from the VQA-X [5] dataset with correctly predicted answers. Then, subjective evaluation is performed by 3 different annotators. Note that each annotator has to select one out of 4 choices: *yes, weak yes, weak no,* and *no*, as a response for whether the explanation justifies the answer. And, these 4 decisions are numerically mapped to 1, 2/3 , 1/3 , and 0, respectively. With averaged results obtained for each method, we present the performance comparisons in the following table. From this table, we see our proposed Rapper is preferable by the users in terms of subjective plausibility and faithfulness assessment. This conclusion also aligns with the objective quantification evaluation of Table 1 presented in the main paper.
> >
> >
> > | Method | Human score regarding plausibility (&uarr;) | Human score regarding faithfulness (&uarr;) |
> > |-|-|-|
> > | NLXGPT [7] | 0.771 | 0.795 |
> > | S3C [8] | 0.797 | 0.811 |
> > | *Rapper* (Ours) | **0.845** | **0.859**  |
> >
> >
> >
> > **Q4: How the proposed RLNF stage tackles the hallucination problem?**
> >
> > **A4:**  We thank the reviewer for giving us the opportunity to clarify our proposed method.
> >
> > Hallucination in VQA refers to the problem when the output is not related to the visual image [9]. With a rationale generator $G$ and a reasoning module $M$ introduced in our Rapper, the proposed RLNF scheme tackles the hallucination problem by enforcing deployed rationale generator $G$ to inject the visual content (of the input image) into the produced rationales. This is achieved by performing the proposed RLNF to optimize the rationale generator $G$ with the reward calculated from the outputs (i.e., answers and explanations) of the pre-trained/freezed reasoning module $M$. The reason why we use the reasoning module output to guide the learing of $G$ is that, generating faithful explanations and accurate answers requires $G$ to input $M$ with rationales associated with proper *visual* clues. Note that, optimization of $G$ is a non-differentiable operation, since the guidance from $M$ cannot be back-propagated to $G$ due to the fact that the rationales from $G$ are text prompts instead of soft features. This is the reason why we propose an RL strategy to perform the above end-to-end training scheme.
> >
> > We also note that, the core idea and learning scheme of our RLNF aligns with the hypothesis from Kadavath, et al. [10], which trains the model towards a faithful one when it exhibits greater confidence in their output answers. As confirmed in Table 2, our Rapper achieved the best performance on RefCLIPScore (i.e, metric for evaluating faithfulness). We thank the reviewer again for giving us the opportunity to clarify how our RLNF tackles hallucination in NLE.
> >
> > [9] Ji, et al., Survey of hallucination in natural language generation. ACM 2023.
> >
> > [10] Kadavath, et al. Language models (mostly) know what they know. arXiv 2022.

---

### Official Review · Reviewer_iCAp · 2023-10-31

**Soundness:** 3 good
**Presentation:** 3 good
**Contribution:** 3 good
**Rating:** 6
**Confidence:** 4

**Summary:**

The authors propose a reinforced rationale-prompted paradigm (Rapper) for natural language explanation (NLE) in visual question answering (VQA). They aim to generate plausible and faithful NLEs to address issues like implausibility and hallucination in existing VQA-NLE methods. Rapper has two stages - knowledge distillation from large language models (LLMs) and reinforcement learning from NLE feedback (RLNF). In stage 1, it elicits pseudo rationales from LLM to encourage plausibility and filters rationales using QA model for quality.
In stage 2, it uses RLNF with answer and explanation scores as rewards to inject visual facts into rationales, improving faithfulness.
RAPPER, when evaluated on VQA-X and e-SNLI-VE datasets, achieves new SOTA on both for NLE metrics. It shows better plausibility via higher CIDEr and SPICE scores compared to prior VQA-NLE methods and demonstrates improved faithfulness through higher RefCLIPScore than previous methods. The approach reduces hallucination and implausibility qualitatively over other approaches.

**Strengths:**

The paper offers a novel two-stage approach to inject both language-based facts and visual content into rationales.
It leverages powerful knowledge and reasoning capabilities of LLMs through distillation. RLNF provides a way to align rationales with visual input for faithfulness. Rationale prompting is interpretable and improves reasoning module's NLE. Training is end-to-end, does not need ground truth rationales.
Moreover, the qualitative results show more precise and reasonable NLEs: if achieves new SOTA on VQA-X and e-SNLI-VE for all NLE metrics in both filtered and unfiltered settings. It also shows higher CIDEr and SPICE scores demonstrating enhanced plausibility of NLEs.
Improved RefCLIPScore indicates increased faithfulness and reduced hallucination.
Ablations validate importance of both knowledge distillation and RLNF stages and analysis of derived rationales indicates progressively better quality.
Qualitative examples exhibit more visually grounded and plausible NLEs than prior methods. It also reduces cases of implausible and hallucinated explanations over other VQA-NLE approaches.
The claims seem reasonably supported by the quantitative and qualitative results on the standard benchmarks. The improved performance across NLE metrics substantiates the effectiveness of the Rapper approach for plausible and faithful explanation generation. The ablation studies validate the contribution of the individual components. The qualitative examples provide additional evidence that Rapper produces more precise and reasonable rationales and explanations.

**Weaknesses:**

Some potential weaknesses include:
The approach relies on eliciting high-quality pseudo rationales from the LLM, but the process for doing so is not extensively analyzed. In fact LLMs, especially smaller ones  (relative to e.g. GPT4) are prone to hallucinations.
The impact of different choices of LLM for knowledge distillation is not addressed.
Evaluation is limited to VQA; extending Rapper to other VL tasks may reveal additional challenges.
More human evaluations on plausibility and faithfulness could further validate the approach.

**Questions:**

How did you determine the optimal hyperparameters (e.g. threshold τ) for filtering pseudo rationales from the LLM? Was any tuning or analysis done to validate these settings?
Did you experiment with different LLMs for knowledge distillation? If so, how did the choice of LLM impact the quality of the distilled rationales?
You mention the potential to extend Rapper to other vision-language tasks. What challenges do you anticipate in adapting the approach to other datasets and tasks?
The elicitation process for pseudo rationales is a key component of your approach but is not analyzed in depth. Can you provide more details on this process and how the prompts were designed?
Can you discuss any trade-offs between plausibility and faithfulness you observed? Does optimizing one tend to hurt the other?

---

> ### Author Response · Authors · 2023-11-23
> **Responses to Reviewer iCAp (1/2)**
>
> **Q1: Choices of LLM for knowledge distillation (KD).**
>
> **A1:** We thank the reviewer for raising this issue. We are glad to conduct additional experiments to assess the LLM choices for KD in our framework.
>
> Instead of using LLaMA [1] in our proposed work, we now replace LLaMA with GPT-3.5 [2] for experiments, as detailed below. To obtain distilled rationales, we consider the rationales produced by GPT-3.5 as pseudo grounth for training the deployed generator $G$. As explained in Q5, such pseudo ground truth rationales are collected via performing in-context learning (ICL) on the pre-trained LLM of GPT-3.5. To assess the quality of such distilled rationales, we take image-question pairs and the generated rationales as the inputs to conduct VQA and calculate its accuracy, following the ablation studies reported in Fig. 4. That is, we apply the pre-trained multimodal large language model of mPlug-Owl [3] to take the image-question pair with the produced rationale to output the answer. Thus, the higher improved accuracy means the higher quality of rationales.
>
> With the above experiments, we observe that further improved accuracy can be achieved (i.e., 69.84% with LLaMA vs. 71.35% with GPT-3.5). This suugests that our proposed framework is applicable to the use of SOTA LLMs if computational environments are allowed. We will be happy to add such suggested experiments and evaluations to Sec 4.3, verifying the robustness of our proposed framework.
>
> [1] Touvron, Hugo, et al. Llama: Open and efficient foundation language models. arXiv 2023.
>
> [2] OpenAI, 2022. Introducing chatgpt.
>
> [3] Ye et al. mplug-owl: Modularization empowers large language models with multimodality." arXiv 2023.
>
>
> **Q2: More human evaluations on plausibility and faithfulness.**
>
> **A2:** We thank the reviewer for suggesting additional experiments for subjective evaluation. Following the evaluation setting/process applied in NLXGPT [4] and S3C [5] (i.e., two SOTAs on VQA-NLE), we randomly select 200 test samples from the VQA-X [6] dataset with correctly predicted answers. Then, subjective evaluation is performed by 3 different annotators. Note that each annotator has to select one out of 4 choices: *yes, weak yes, weak no*, and *no*, as a response to whether the explanation justifies the answer. And, these 4 decisions are numerically mapped to 1, 2/3, 1/3, and 0, respectively. With averaged results obtained for each method, we present the performance comparisons in the following table. From this table, we see our proposed Rapper is preferable by the users in terms of subjective plausibility and faithfulness assessment. This conclusion also aligns with the objective quantification evaluation of Table 1 presented in the main paper.
>
> | Method | Human score regarding plausibility (&uarr;) | Human score regarding faithfulness (&uarr;) |
> |-|-|-|
> | NLXGPT [4] | 0.771 | 0.795 |
> | S3C [5] | 0.797 | 0.811 |
> | *Rapper* (Ours) | **0.845** | **0.859**  |
>
> [4] Sammani et al. Nlx-gpt: A model for natural language explanations in vision and vision-language tasks. CVPR 2022.
>
> [5] Suo et al. S3C: Semi-Supervised VQA Natural Language Explanation via Self-Critical Learning. CVPR 2023.
>
> [6] Park et al. Multimodal explanations: Justifying decisions and pointing to the evidence. CVPR 2018.
>
> **Q3: How to determine the optimal hyperparameters (e.g. threshold τ) for filtering pseudo rationales from the LLM?**
>
> **A3:** Following [7], we simply set the threshold τ for filtering pseudo rationales as 0.54. As for other hyperparameters such as the RL discount factor for reward $\gamma$ and entropy weight $\alpha$ for calculating the reward function by Eqn (9), we follow [8] and set $\gamma$ = 1.0 and $\alpha$ = 0.2.
>
> [7] Changpinyo et al. All you may need for vqa are image captions. NAACL 2022.
>
> [8] Ziegler et al. Fine-Tuning Language Models from Human Preferences. arXiv 2019.

---

> > ### Author Response · Authors · 2023-11-23
> > **Responses to Reviewer iCAp (2/2)**
> >
> > **Q4: Evaluation is limited to VQA. What challenges do you anticipate in adapting the approach to other VL datasets and tasks?**
> >
> > **A4:** We thank the reviewer for the suggestion. We are happy to point out that we did apply our Rapper on another VL-NLE (natural lange explanation) task of Visual Entailment (VE), as listed in the lower part of Table 1 of our main paper.
> >
> > We note that, VE requires one to assign a label determining the relationship between a premise image and a textual hypothesis (i.e., “entailment”, “neutral”, and “contradiction”), while giving an explanation for supporting the aforementioned relationship [9]. And, VE is tyically applied in cross-modal retrieval, etc. applications. In Table 1 of our main paper, we report the performance on the **e-SNLI-VE** benchmark, in which we observed that our Rapper performed favorably against NLXGPT [4] and S3C [5] in terms of different NLG metrics (i.e., CIDEr, SPICE, and ROUGE-L). We will add the definition of VE with the above additional clarification to Sect. 4.1 of our main paper.
> >
> > While the above extension confirms the use of our Rapper for other VL-NLE tasks, one can simply apply existing VLMs [10] with a supervised learning setting for standard VQA tasks (e.g., VQAv2 [11] simply requires one to answer a question from an input image without giving any desirable explanation). In other words, it would be preferable to apply our Rapper to the VL tasks which require NLE as parts of the outputs. However, while learing of Rapper does not observe any ground truth rationales, ground truth output explanation is required as its training supervision. This will be the current limitation of our framework.
> >
> > [9] Do et al. e-snli-ve: Corrected visual-textual entailment with natural language explanations. CVPRW 2020.
> >
> > [10] Liu et al. Improved baselines with visual instruction tuning. arXiv 2023.
> >
> > [11] Goyal et al. Making the V in VQA Matter: Elevating the Role of Image Understanding in Visual Question Answering. CVPR 2017
> >
> >
> >
> >
> > **Q5: The elicitation process for pseudo rationales is a key component but not analyzed in depth. More details on this process are desirable.**
> >
> > **A5:** We thank the reviewer for pointing this out. While the task description and few-shot demonstrations have been presented in Supplementary A.4, we are glad to clarify this issue as follows.
> >
> > The elicitation process of pseudo rationales is using in-context learning (ICL) to prompt LLaMA. For VQA-X [6] in our experiments, the input includes a task description with four randomly selected few-shot demonstrations. Take the following task description as an example:
> >
> > *“Please gives the rationale that justifies the following explanation and answer, according to the question.”*
> >
> > A demonstration we select is presented below:
> >
> > *“question: does this kid look excited
> > answer: yes
> > explanation: the boy has his arms in the air and a smile on his face
> > rationale: the rationale is that arms in the air and a smile are indications of excitement”*
> >
> > With the above formats, we are able to perform ICL to generate rationales for further processing and learning. Please refer to Supp. A.4 for more examples and demonstrations.
> >
> >
> > **Q6: Any trade-offs between model plausibility and faithfulness?**
> >
> > **A6:** We thank the reviewer for raising this concern, and we are more than happy to provide the following clarification. There is no trade-off between plausibility and faithfulness in Rapper. As depicted in Fig. 2, our Rapper is a two-stage learning scheme. The first stage is a knowledge distillation (KD) process from LLM which focuses on generating plausible NLE. Based on produced plausible NLE outputs, the second stage performs reinforcement learning from NLE feedback (RLNF), which is designed to improve the output faithfulness. This RL technique ensures the NLE generation process reflects and corresponds to the visual content presented in the input image. From the above design, it can be seen that there is no trade-off between the two desirable properties.
> >
> > As illustrated in the upper part of Table 3, Rapper w/o RLNF (i.e., the baseline model w/ only the KD stage) improves the the plausibility metrics (in CIDEr and SPICE) when compared to the baseline model (i.e., Rapper w/o RLNF w/o KD). With further employment of RLNF, the full version of Rapper improves Rapper w/o RLNF from 65.66 to 67.05 in terms of RefCLIPScore (i.e., a metric to evaluate faithfulness in NLE), as shown in Table 2. These results confirm that our two designed stages for introducing the two desirable properties to our learned model.

---

### Official Review · Reviewer_EbLA · 2023-11-04

**Soundness:** 3 good
**Presentation:** 3 good
**Contribution:** 2 fair
**Rating:** 3
**Confidence:** 3

**Summary:**

The paper is about mitigating implausibility and hallucination problems (non-informative or contradicting visual context) for generating natural language explanation (NLE) under VQA problems. To mitigate the issue, the authors introduced a notion of “rationale” which is similar to chain-of-thought prompting. To combat the issue of generating rationale without training data, the authors distill rationale from LLMs into the rationale generator. To penalize hallucinated rationale, “Reinforcement Learning from NLE Feedback” is used. The combination of proposed method brings a marginal improvement on benchmarks.

**Strengths:**

- The paper is largely well-written and easy to understand.
-  The function of each component in the method is clear and sound.
- The method achieves SOTA on NLE benchmarks.

**Weaknesses:**

- The role of using rationale to improve implausibility and hallucination is unclear. It is well-known that chain-of-thought can improve reasoning. However, it is unclear to me if adding one more step, i.e., rationale could really mitigate hallucination.
- While the method is sound, I’m not very convinced that we cannot just use large vision language models and perform a chain-of-thought style prompting (which was actually the inspiration of this method)? How does large vision language models (e.g. BLIP-family or LLAVA models) perform?
- In ablation study table, the impact of proposed method is small. Especially, RLNF effect is small.
- A clear definition of “rationale” is not presented in the paper. Only mentioned that it is like an “intermediate” just like in chain-of-thought prompting.

**Questions:**

- Please answer my questions in “weaknesses” section. I may raise score if the rebuttal is satisfactory.

---

> ### Author Response · Authors · 2023-11-15
> **Responses to Reviewer EbLA (1/2)**
>
> **Q1: It is well-known that chain-of-thought (CoT) can improve reasoning. The role of using rationale to improve implausibility and hallucination is unclear. Why generating rationale could mitigate implausibility/hallucination?**
>
> **A1:** We thank the reviewer for giving us the opportunity to clarify the raised issues.
>
> We first explain **how our work is fundamentally different from CoT promoting [1].** In NLP, the use of CoT has been a common technique, which utilizes a pre-trained LLM with manually-designed prompts with in-context learning (ICL) techniques, so that additional reasoning descriptions can be produced by that LLM. However, since the above technique is designed for language data, extending CoT to vision and language is not a trivial task. More specifically, it is not clear how to determine prompting conseutive examples which allow pre-trained vision-language models like LLAVA [2] to fully extract the information from both visual and text modality data for vision-language explanation tasks [3]. Another concern is that, since CoT (for NLP) requires manually selected prompts and does not provide additional facts related to the output, the output would suffer from hallucination [4]. For our Rapper, the deployed rationale generator is designed to exploit language-based facts and visual content from cross-modality data inputs (i.e., with our learning schemes via knowledge distillation (KD) and Reinforcement Learning from NLE Feedback (RLNF), as explained in the following paragraph). With the learned rationales as the text prompts, plausible and faithful natural language explanations (NLEs) for VQA can be performed. Based on the above explanations, we hope the reviewer could see **why CoT cannot be directly applied for reasoning in vision-language tasks.**
>
> We now explain **how rationale learning helps mitigate implausibility and hallucination problems**. As noted in our paper, *implausibility* refers to the problem when NLEs are irrelevant to the questions or contradictory to the established supporting facts [5]. As discussed in Sect. 3.2 (Plausible NLE Generation) and depicted in Fig. 2(A), we tackle this problem by learning rationales enriched with language-based facts from a pre-trained LLM. That is, we deploy a rationale generator $G$ in Rapper, which is trained via distillating the knowledge from LLM. With such rationales with language-based facts serving as additional text prompts, together with the image-question pair as the inputs, the subsequent reasoning module $M$ can be trained to provide answers with plausible explanations.
>
> On the other hand, *hallucination* refers to the problem when the output explanation is not related to the visual image [6]. As noted in Sect. 3.3 (Faithful NLE Generation) and illustrated in Fig. 2(B), this is tackled by the proposed Reinforcement Learning from NLE Feedback (RLNF) technique, which jointly trains $G$ and $M$ for enforcing the learned rationale to fully exploit the visual content. The motivation of our RLNF aligns with the hypothesis of Kadavath, et al. [7], which views a learning model towards a faithful one when it exhibits greater confidence in their output answers. In other words, with the proposed RL scheme and objective rewards, the outputs of our reasoning module $M$ need to be supported by the rationale (i.e., predticed by the generator $G$) containing the visual content from the input image. As confirmed in Table 2, our Rapper achieved the best performance on faithful NLE for VQA in terms of RefCLIPScore. We thank the reviewer again for giving us the opportunity to clarify how our Rapper mitigates the problems of implausibility and hallucination.
>
>
> [1] Wei et al. Chain-of-thought prompting elicits reasoning in large language models. NeurIPS 2022.
>
> [2] Liu et al. Visual instruction tuning. NeurIPS 2023.
>
> [3] Li et al. Mimic-it: Multi-modal in-context instruction tuning. arXiv 2023.
>
> [4] Dhuliawala et al. Chain-of-verification reduces hallucination in large language models., arXiv 2023.
>
> [5] Majumder et al., Knowledge-grounded self-rationalization via extractive and natural language explanations. ICML 2022.
>
> [6] Ji et al. Survey of hallucination in natural language generation. ACM 2023.
>
> [7] Kadavath et al. Language models (mostly) know what they know., arXiv 2022.

---

> ### Author Response · Authors · 2023-11-15
> **Responses to Reviewer EbLA (2/2)**
>
> **Q2: While the method is sound, why not just use large vision-language models and perform a CoT style prompting (which inspires this method)? How do large vision language models (e.g., BLIP-family or LLAVA models) perform?**
>
> **A2:** We thank the reviewer for the positive feedback. We are glad to perform additional experiments as suggested. As noted in our response to Q1, we would like to point out that existing VL models like LLAVA [2] or BLIP-family [8, 9] do *not* support in-context learning (ICL). Nevertheless, we are happy to take the suggestion from the reviewer to assess the use of VL models with properly designed ICL. We now consider the VQA-X dataset and compare Rapper with Otter [10] (i.e., a VL model built on LLaMA7B). For Otter, we follow the ICL setting in PICa [11] to random sample 16 samples as the demonstration prompts for ICL.
>
> * Filtered results on the VQA-X dataset
>
> |Method|Bleu4|Meteor|Rough-L|CIDEr|SPICE|RefCLIPScore|Acc.|
> |-|-|-|-|-|-|-|-|
> |Otter w/ ICL|4.9|14.9|26.7|29.0|13.0|63.68|39.6|
> |*Rapper* (ours)|**31.8**|**24.3**|**52.9**|**124.0**|**24.5**|**67.05**|**87.3**|
>
> * Unfiltered results on the VQA-X dataset
>
> | Method|Bleu4|Meteor|Rough-L|CIDEr|SPICE|
> |-|-|-|-|-|-|
> | Otter w/ ICL|4.5|12.5|24.9|22.0|9.7|
> | *Rapper* (ours)|**30.0**|**23.3**|**51.3**|**116.0**|**23.2**|
>
> From the above tables, we see that Otter with ICL did not achieve comparable explanation performances as ours for both filtered and unfiltered cases (i.e., explanations associated with correct answers or not). This suggests that simple combination of VL models with ICL would not sufficiently address the VQA with natural language explanation, and a properly design VL model like ours would be desirable.
>
> [8] Li et al. Blip-2: Bootstrapping language-image pre-training with frozen image encoders and large language models. ICML 2023.
>
> [9] Dai et al. InstructBLIP: Towards general-purpose vision-language models with instruction tuning. NeurIPS 2023.
>
> [10] Li et al. Otter: A multi-modal model with in-context instruction tuning. arXiv 2023.
>
> [11] Yang et al. An empirical study of gpt-3 for few-shot knowledge-based vqa. AAAI 2022
>
> **Q3: Small impact of the proposed method in the ablation study table (e.g., w/ or w/o RLNF)**
>
> **A3:** We thank the reviewer for pointing this out, and we are glad to clarify the performance improvements. In the upper part of Table 3, we perform ablation studies to verify the effectiveness of KD and RLNF. From this table, we see that the introduction of KD produced improved performance over the baseline model (e.g.,  CIDEr: 112.1 -> 120.2 and SPICE: 22.3 -> 24.2). This confirms the use of KD for predicting plausible outputs. Since our RLNF is deployed to mitigate the hallucination problem **based on plausible prediction**, adding RLNF to KD only increased the performance with smaller margins (e.g., CIDER: 120.2 -> 124.0 and SPICE: 24.2 -> 24.5). However, such improvements are expected. Recall that, as we explained in Q1, our Rapper is based on the introduction of KD and RLNF for providing plausible and faithful language explanation, and we cannot simply utilize RLNF without enforcing the NLE to be plausible.
>
> Nevertheless, even with simply KD introduced and without RLNF, our model already performed favorably against SOTAs like S3C [12] and NLXGPT [13] (e.g., list CIDER/SPICE numbers, as shown in Table 1). And the full version of Rapper (i.e., with KD and RLNF) achieved the best results, which confirms our proposed learning schemes.
>
> [12] Suo et al. S3C: Semi-Supervised VQA Natural Language Explanation via Self-Critical Learning. CVPR 2023.
>
> [13] Sammani et al. Nlx-gpt: A model for natural language explanations in vision and vision-language tasks. CVPR 2022.
>
> **Q4: A clear definition of “rationale” is not presented in the paper. Only mentioned that it is like an “intermediate” just like in chain-of-thought prompting.**
>
> **A4:** We thank the reviewer for the suggestion. Although rationale-based prompting techniques have been presented in [14, 15], a proper definition should be given in our paper for clarity purposes. More precisely, in our work, rationales are the text prompts that generated by the rationale generator $G$, which are injected with language-based facts and visual contents. As illustrated in Fig. 1, the reasoning module $M$ takes the learned rationales as the text prompts, together with the image and question inputs, realizing plausible and faithful NLE. We will add the above definition to the fourth paragraph of the introduction.
>
> [14] Zhang et al. Multimodal chain-of-thought reasoning in language models. arXiv 2023.
>
> [15] Krishna et al. Post hoc explanations of language models can improve language models. arXiv 2023.

---

### Author Response · Authors · 2023-11-23
**General Response**

We sincerely appreciate the valuable time and insightful feedback provided by the reviewers. We are grateful for the opportunity to address the concerns raised by each reviewer, which fundamentally strengthened our work. The strengths pointed out by the reviewers include:

1. **(Novelty)**: The proposed framework for tackling implausibility and hallucination problems in NLE is novel [Reviewer iCAp] and noteworthy [Reviewer iXW4].
2. **(Experiment/Performance)**: Ablated quantitative and qualitative experiments validate the contribution of the individual component [Reviewer iCAp]. SOTA performance supports the effectiveness of the proposed method. [Reviewers EbLA, iCAp]
3. **(Presentation)**: The paper is largely well-written and well-organized. [Reviewers EbLA, iXW4]

We would like to point out that particular concerns are raised, as listed below. Please refer to the responses to each reviewer for further details.
1. How does rationale generation in VQA+explanation differ from **Chain-of-thought (CoT)** in NLP tasks? Why not simply use CoT for vision-language models? [Reviewer EbLA]
2. Why/how does generating rationales address **implausibility and hallucination**? [Reviewers EbLA, iXW4]
3. **Marginal improvements** in ablation studies & proper definition of rationale [Reviewer EbLA]
4. Additional **human studies** & extension to other VL tasks [Reviewers iCAp, iXW4]
5. **Trade-off** between plausibility & faithfulness [Reviewer iCAp]

We thank the reviewers again for the suggestions and the raised issues. Given the recognized strengths in the initial reviews, together with additional experiments and clarification provided during rebuttal, we hope this work would be of great value to vision-language research communities, and the reviewers will be able to provide proper evaluation during the next phase.

---

### Meta-Review · Area_Chair_WxTg · 2023-12-14

**Metareview:**

While the work shows promise in generating more plausible and faithful VQA explanations, the reviewers identify several limitations and request further clarification before fully endorsing the work. I recommend for a weak accept with the expectation of addressing the identified weaknesses.

Strengths include combining knowledge distillation and reinforcement learning to inject visual and factual data into rationales.  Improved RefCLIPScore suggests less hallucination and increased faithfulness. Qualitative results demonstrate more precise and visually grounded rationales compared to prior methods.

However, the process for eliciting high-quality rationales from the LLM is unclear and potentially prone to hallucinations, especially with smaller LLMs. Evaluation focuses mainly on VQA, raising concerns about generalizability to other vision-language tasks. Human evaluation is provided during discussion phase, but need further cross-checking on human experimental setup details.

**Justification For Why Not Higher Score:**

To improve, the authors need to provide further evidence for the generalizability and human-perceived effectiveness of RAPPER's explanations, and the human evaluation details are needed in the final draft.

**Justification For Why Not Lower Score:**

I am recommending an acceptance but I wouldn't mind if the paper gets rejected, due to limited analysis, lack of human study details, limited evaluation scores and tasks (mainly on VQA), etc.

---

### Decision · Program_Chairs · 2024-01-16

Accept (poster)